# From Infection to Malignancy: Tracing the Impact of Human Papillomavirus on Uterine Endometrial Cancer in a Nationwide Population-Based Cohort Study

**DOI:** 10.3390/v15122314

**Published:** 2023-11-25

**Authors:** Pei-Ju Wu, Stella Chin-Shaw Tsai, Jing-Yang Huang, Maw-Sheng Lee, Po-Hui Wang, Frank Cheau-Feng Lin

**Affiliations:** 1Institute of Medicine, Chung Shan Medical University, Taichung 40201, Taiwan; mushroom307@gmail.com (P.-J.W.); cshe961@csh.org.tw (J.-Y.H.); 2Department of Obstetrics and Gynecology, Chung Shan Medical University Hospital, Taichung 40201, Taiwan; msleephd@gmail.com; 3Superintendent Office, Tungs’ Taichung MetroHarbor Hospital, Taichung 43503, Taiwan; tsaistella111@nchu.edu.tw; 4Department of Post-Baccalaureate Medicine, National Chung Hsing University, Taichung 40227, Taiwan; 5Department of Medical Research, Chung Shan Medical University Hospital, Taichung 40201, Taiwan; 6Lee Women’s Hospital, Taichung 40652, Taiwan; 7School of Medicine, Chung Shan Medical University, Taichung 40201, Taiwan; 8Department of Surgery, Chung Shan Medical University Hospital, Taichung 40201, Taiwan

**Keywords:** cancer epidemiology, cohort, endometrial cancer, human papillomavirus, population-based, uterine corpus cancer

## Abstract

Uterine endometrial cancer (EC) is the most common gynecological malignancy in Taiwan. This study aimed to investigate the association between human papillomavirus (HPV) infection and the development of uterine EC among Taiwanese women. A nationwide population cohort research approach was employed, leveraging longitudinal health insurance databases (LHID 2007 and 2015) from the National Health Insurance Research Database alongside data from the Taiwan Cancer Registry datasets. A comparative analysis examined 472,420 female patients with HPV infection and 944,840 without HPV infection. The results demonstrated that the HPV cohort exhibited a significantly elevated risk of uterine EC, as evidenced by an adjusted hazard ratio (aHR) of 1.588 (95% CI: 1.335–1.888). Furthermore, this elevated risk extended to type 1 EC with an aHR of 1.671 (95% CI: 1.376–2.029), specifically the endometrioid adenocarcinoma subtype with an aHR 1.686 (95% CI: 1.377–2.065). Importantly, these findings were statistically significant (*p* < 0.001). In conclusion, this research unveils a potential association between HPV infection and an increased risk of uterine EC, particularly the type 1 endometrial cancer subtype, within the Taiwanese female population. These findings have implications for preventive measures and screening programs targeting HPV infection to reduce the risk of this prevalent gynecological malignancy in Taiwan.

## 1. Introduction

According to a World Health Organization report, in 2020, 8.1 million people worldwide were diagnosed with cancer and 9.6 million died of cancer [1]. In 2020, the Taiwan Health Promotion Administration of the Ministry of Health and Welfare documented cancer as a prominent contributor to global mortality and a significant cause of death within Taiwan. The reported mortality rate was 115.94 deaths per 100,000 person-years [2]. Mutations leading to cancer may be related to the interaction between environmental and personal genetic factors. Among these carcinogens, viral infection has been proven to be a significant risk [1]. Human papillomavirus (HPV) is a prevalent sexually transmitted infection affecting nearly all sexually active individuals at some point in their lives [3,4]. Extensive research has established HPV as a crucial factor in the development of invasive cervical cancer (CC) globally [5]. Moreover, HPV has been linked to other malignancies within the female reproductive system, such as vulvar and vaginal cancers [6]. Considering the close anatomical relationship to the cervix, HPV could invade the endometrium through ascending infections and potentially play a role in the etiology of endometrial cancer (EC) of the uterine corpus [7].

Endometrial cancer (EC) is a prevalent malignancy affecting over 300,000 women globally each year, with a higher incidence in more developed countries [7], and stands as the most common gynecological malignancy in Taiwan [8]. The 2020 age-standardized incidence rate of uterine corpus cancer was 16.30 per 100,000 person-years of female individuals. Among these cases, EC constitutes the majority, and it is traditionally categorized into two main types, namely, type 1 and type 2, based on histological, clinical, and metabolic characteristics [9]. Endometrioid endometrial cancer [10], which belongs to type 1, constitutes the majority and accounts for more than 80% of EC [11]. In contrast, type 2 EC includes serous endometrial cancer, clear cell endometrial cancer, and carcinosarcoma, making up approximately 10%, 3%, and less than 2% of cases, respectively. However, it is essential to note that some instances exhibit overlapping characteristics between type 1 and type 2 EC. Previous research has shown that 10–19% of endometroid ECs are high-grade and have more comparable behaviors to type 2 EC in terms of clinical, histopathological, and molecular features. These findings imply that there are still several etiologic differences between type 1 and type 2 EC [12].

Some prior studies have investigated the prevalence of HPV in EC. However, the reported prevalence rates have shown significant variations, ranging from 0% to 61.1%, indicating substantial heterogeneity between these studies [7]. Since 1995, Taiwan has implemented a single-payer national health insurance system, providing coverage to approximately 99% of its population. Furthermore, all cancer patients in Taiwan must be registered in the cancer registry database maintained by the National Institution of Health Promotion. A nationwide population-based cohort study is regarded as a reliable method to evaluate cancer etiology [13,14,15]. This study investigated the relationship between HPV and uterine corpus cancer, especially endometrial cancer, among Taiwanese women. As cervical cancer has a well-established causal link with HPV, examining both CC and EC allows us to validate the reliability of our HPV cohort identification method against this “positive control” before investigating the novel association with endometrial cancer.

## 2. Materials and Methods

### 2.1. Study Design and Data Source

This study explored the hazard ratio (HR) of uterine corpus cancer in patients with HPV infection by using a retrospective nationwide cohort design. Data were collected from Taiwan’s National Health Insurance Research Database (NHIRD), which consists of claims data from all of Taiwan’s hospitals and clinics, dating from 1995, and presently covers more than 99% of the Taiwanese population. In addition, the Taiwan Cancer Registry (TCR) datasets, a population-based cancer registry system established in 1979, were used to obtain detailed information on cancer diagnosis, incidence, patient demographics, date of diagnosis, tumor site, histology, stage, and the evaluation of cancer control and prevention. NHIRD and TCR are cross-linked and regulated by the Taiwan Health and Welfare Data Science Center. The study was approved by the Institutional Review Board of Chung Shan Medical University Hospital, Taichung (#CS17129). Only cases that both NHIRD and TCR recognized as cancer were indicated as cancer.

### 2.2. Identification of Study Cohort and Exclusion Criteria

The NHIRD included about 26 million unique persons between 2007 and 2015 (Figure 1). Initially, we randomly sampled 4.7 million women (about 1/5 of the total population) for analysis. HPV infection was identified using the International Classification of Diseases Clinical Modification (ICD-9-CM) codes 079.4, 078.1, 078.10078.12, 078.19, 759.05, 795.09, 795.15, 795.19, 796.75, and 796.79 (Table 1). Of the HPV infection population, we excluded (1) individuals diagnosed with HPV before 2008 due to the left-truncation of data and (2) individuals diagnosed with any cancer (including uterine cancer) before diagnosis of HPV infection or index date. After exclusion, the remaining female individuals without HPV were individually matched with those with HPV by age on the index date (the date of first diagnosis of HPV infection).

### 2.3. Study Events, Covariates, and Primary Outcomes

The occurrence of either cervical cancer or uterine corpus cancer was determined using the TCR datasets for each study individual from the index date until the end of the study (31 December 2017). Diagnosing malignant neoplasm of the ne cervix was defined as ICD10 code C53. Diagnosing malignant neoplasm of the uterine corpus (endometrial cancer) was defined as ICD10 codes C54 and C55. In addition, the ICD-O-3 morphology code was included in the TCR. The risks of type 1 and type 2 endometrial cancers were further evaluated for each study individual. We identified the following morphology codes as (1) type 1 endometrial cancer: 814, 856, 801, 838, and 848; and (2) type 2 endometrial cancer: 857, 802, 831, 844, 898, 895, 826, 805, 845, and 846 (Table 1).

Demographic data were also compared, including urbanization, residential area, and income status. The community’s urbanization level was stratified into seven classifications (1 being the most urbanized and 7 being the least). Liu and colleagues introduced this classification system at the Taiwan National Health Research Institute, drawing from the 2000 Taiwan census data [16].

Furthermore, comorbidities were determined based on ICD-9-CM codes. They were confirmed if there were at least two ICD-9-CM records for outpatient visits or one diagnostic code for hospitalization during the baseline period (within one year before the index date). The definitions of exposure, study event, and comorbidity are listed in Table 1. The primary outcome was to compare the risk of cervical cancer and uterine corpus cancer, including endometrial cancer type 1 and type 2, among patients with HPV infection.

### 2.4. Statistical Analysis

The incidence rate (per 100,000 person-years) and its 95% confidence interval (95% CI) were estimated using Poisson regression. A 2-tailed Student’s *t*-test was used to compare the mean difference between continuous variables, while the chi-square test was used for categorical variables. For the time-to-event analysis of the longitudinal follow-up, the event was defined as the date of cancer onset. Follow-up was censored when a patient died or cancelled insurance. The Kaplan–Meier estimator was used to calculate the cumulative incidence probability of CC and EC among the study groups and tested with log rank. A multiple Cox regression model was conducted to calculate CC and EC’s adjusted hazard ratio (aHR) after adjusting the covariates, including age at index date, urbanization, residential area, economic status, and comorbidities. Statistical significance was defined as *p* < 0.05. SAS V. 9.4 software (SAS Institute Inc., Cary, NC, USA) was used for all statistical analyses.

## 3. Results

### 3.1. Study Population Examining Uterine Body and Cervical Cancer Incidences in Correlation with HPV Infection

In total, 554,926 patients were diagnosed with HPV between 2007 and 2015, with an incidence rate of 11.7%. Of the HPV infection population, (1) 69,112 female individuals were diagnosed with HPV before 2008, and (2) 13,394 patients diagnosed with any cancer (including uterine cancer) before diagnosis of HPV infection were excluded. Finally, the study matched 472,420 HPV and 944,840 non-HPV patient cohorts to evaluate uterine cancer risk (Figure 1).

### 3.2. Demographic Characteristics of HPV-Infected and Noninfected Individuals

Table 2 shows the essential demographic characteristics of the study population. Individuals with HPV infection exhibited several distinct features. They were more likely to reside in urban areas, comprising 65.52% of the population in groups 1 and 2 of urbanization. Moreover, this group had a lower percentage of individuals with a low income, accounting for only 0.94%. Additionally, individuals with HPV infection had a higher prevalence of specific comorbidities, including ischemic heart disease, hypertension, hyperlipidemia, and chronic obstructive pulmonary disease (COPD). Furthermore, they were more prone to peptic ulcers, gastrointestinal (GI) bleeding, chronic kidney diseases, and gout. Conversely, women with HPV infection showed a lower prevalence of diabetes mellitus and abnormal liver function. Furthermore, they were less inclined to experience renal failure.

### 3.3. Incidence of Uterine and Cervical Cancer among the HPV-Exposed and Unexposed Individuals

Table 3 shows the events and incidence rates for CC and EC (including type 1 and type 2) among HPV and non-HPV cohorts and the hazard ratios (HRs). The incidence rates of uterine CC were 41.65 (95% CI: 38.82–44.68)/100,000 person-years in the HPV group and 18.98 (95% CI: 17.63–20.42)/100,000 person-years in the non-HPV group, with an adjusted HR of 2.225 (2.008–2.464) *p* < 0.001 (Figure 2A). The incidence rates of uterine EC were 12.60 (95% CI: 11.08–14.31)/100,000 person-years in the HPV group and 7.81 (95% CI: 6.96–8.76)/100,000 person-years in the non-HPV group; the adjusted HR was 1.588 (95% CI: 1.335–1.888), *p* < 0.001 (Figure 2B). Specifically, the incidence rate of type 1 EC was 0.86 (95% CI: 0.74–0.99)/100,000 person-years in the HPV group and 0.50 (95% CI: 0.44–0.57)/100,000 person-years in the non-HPV group, with an adjusted HR 1.671 (95% CI: 1.376–2.029), *p* < 0.001 (Figure 2C); the incidence rates of type 2 EC were 0.08 (95% CI: 0.05–0.13)/100,000 person-years in the HPV group and 0.06 (95% CI: 0.04–0.08))/100,000 person-years in the non-HPV group, HR, 1.450 (95% CI 0.791–2.656), without statistical significance. Women with HPV infection did not exhibit an increased risk of having type 2 EC. Thus, women with HPV infection showed an increased risk of uterine corpus cancer and type 1 EC.

### 3.4. Factors Associated with Uterine Endometrial and Cervical Cancers

Table 4 shows the multivariate analyses with Cox regression analysis for uterine CC and EC. The results showed that women with HPV infection exhibited a higher risk of CC than non-HPV women. Other independent risk factors for cervical cancer were age, abnormal liver function, and renal failure. In addition, other independent risk factors for uterine EC were age, hypertension without COPD, and gout. Among these, age is a crucial independent risk factor, with the peak at 40–60 years for both EC (aHR, 6.547; 95% CI: 5.035–8.514) and CC (aHR, 1.124; 95% CI: 1.001–1.126).

## 4. Discussion

Our study revealed a link between HPV infection and elevated risk of uterine cervical cancer among women, providing additional population-level data that align with existing evidence supporting the role of HPV as a key carcinogenic contributor to cervical cancer [17]. An increased risk of uterine EC was found in women with HPV infection in our study. Among the uterine ECs, the risk of type 1 EC was elevated in the HPV group with an aHR of 1.671 (95% CI: 1.376–2.029). In addition, this trend was also found in endometrioid endometrial cancer, the most popular type I EC, with an aHR of 1.686 (95% CI: 1.377–2.065). In contrast, the risk of type 2 EC was not demonstrated to be significantly increased in the HPV group, though only a few cases could be collected because of the lower incidence compared to that of type 1 endometrioid endometrial cancer.

As a more developed country, Taiwan has experienced a transition in cancer patterns, moving from CC to EC [2,18], paralleling the shift from squamous cell carcinoma of the lung in smokers to adenocarcinoma in non-smokers. This transition was previously suggested to be linked to HPV infection in our prior study [2,19,20]. In 1981, cervical cancer was the leading cause of female cancer-related mortality in Taiwan, with an incidence rate of 13.5 per 100,000 person-years. However, by 2020, it had descended to the eighth position, with an incidence rate of 3.06 per 100,000 person-years. Interestingly, uterine endometrial cancer, which held the 24th position in 1981 with an incidence rate of 0.3 per 100,000 person-years, had risen to the 11th position by 2020, with an incidence rate of 1.5 per 100,000 person-years. This shift toward EC was particularly prominent in developed areas [7]. The transition from CC to EC suggests a change in the dominant risk factors among women in Taiwan. Traditionally, CC has been associated with factors like HPV infection, sexual behaviors, and cervical screening practices [21]. In contrast, the rise in EC may signal evolving risk factors, such as hormonal influences, obesity, and metabolic factors [22]. Preventive measures that were initially aimed at reducing CC risk, such as HPV vaccinations and cervical screenings, may need to be complemented with strategies to address the changing landscape of gynecological cancers. This includes promoting healthier lifestyles, addressing obesity, and fostering awareness about EC risk factors.

Our study revealed that the prevalence of HPV was higher in urban areas and among high-income individuals (Table 2). Additionally, we identified specific comorbidities associated with HPV infection, including ischemic heart disease, hypertension, and hyperlipidemia, with hypertension and gout being more prominent among individuals with EC (Table 4). These cardiovascular and metabolic diseases were discernible in individuals from higher socioeconomic backgrounds [23]. Observing a similar pattern of HPV and EC in higher socioeconomic and more developed areas raises intriguing questions about the potential underlying factors that may link these two trends. Individuals in higher socioeconomic brackets may have different health behaviors, including better healthcare-seeking behaviors and healthier lifestyles, which could impact HPV infection rates and the incidence of EC. More developed areas often have well-established healthcare infrastructure, including robust cancer screening programs, providing unique research opportunities. Researchers can conduct more extensive studies and gather data that may help unravel the complex interplay between HPV, EC, and socioeconomic factors.

The ratio of type 1 and type 2 EC is about 10:1 in our study, which is also in agreement with the epidemiologic reports [11]. The peak age range for HPV infection typically occurs between 20 and 40 years of age. Remarkably, this age range aligns with the peak incidence of CC and EC, predominantly affecting individuals between 40 and 60 years old. These age patterns are consistent with the statistical data, emphasizing the reliability and significance of these findings [24,25,26]. This provides further evidence that the age-specific trends in HPV-related cancers are not unique to this study but are reflective of broader population dynamics. The observed age pattern also hints at a relatively long latency period between HPV exposure and the manifestation of cancer. This extended time frame suggests that preventive measures targeting HPV infections and early interventions for high-risk individuals could be vital in reducing the burden of both CC and EC.

HPV is linked not only to vulvar and vaginal cancers within the female reproductive tract [6] but also to a widening spectrum of other cancers, including anogenital, oropharyngeal, and lung cancer, as reported in an increasing body of research [20]. Interest among researchers has grown regarding the potential role of HPV in EC, another prevalent gynecological cancer. Given the close anatomical proximity to the cervix, there is a plausible pathway for HPV to infiltrate the endometrium through ascending infections, potentially contributing to the development of EC [7].

The etiologic association of HPV with the development of CC, both squamous cell carcinoma and adenocarcinoma, has been suggested based on epidemiological studies and molecular technology [27]. Past investigations have established the critical role of HPV’s E6 protein in blocking the function of the p53 protein, thereby suppressing its ability to inhibit the cell cycle and promote apoptosis [28]. Simultaneously, HPV’s E7 protein interferes with the RB gene, releasing E2F and allowing cell cycle progression. E7 binds to cyclin-dependent kinases and active cyclin complexes, further promoting cell cycle progression [5,29]. Interestingly, in the spinous cells adjacent to the basal cells, the production of p21cip1 is stimulated by p53 in response to E7’s actions on cyclin-dependent kinases. However, due to the blockage of p53, p21cip1 expression is diminished in the basal cells, contributing to altered cell cycle regulation.

Previous studies have attempted to investigate the presence of HPV DNA in tumor tissue from endometrial cancer. However, the reported prevalence has shown considerable variability, ranging from 0% to 61.1% [7]. In a prior meta-analysis, the pooled HPV prevalence was 8.5%, notably lower than that observed in other tumors within the female reproductive tract. The considerable heterogeneity of their data may be due to the varied HPV DNA detection methods used in different studies. Nevertheless, these detection methods make it difficult to confirm the integration of HPV into the cellular genome, which is the current basis of carcinogenesis related to HPV infection [30].

Extensive research has illuminated distinct genetic alterations in type 1 and 2 EC. Type 1 tumors are linked to genetic changes involving PTEN, KRAS, CTNNB1, and PIK3CA genes. Additionally, hypermethylation of the MLH1 promoter and microsatellite instability are the prominent features associated with type 1 EC [31]. Prior reports indicate that phosphatase and tensin homolog (PTEN), a tumor suppressor, play a key role in type 1 EC tumorigenesis, and the mutation occurs early in the neoplastic process [32]. EC carcinogenesis was identified as recurrent translocations involving genes in several pathways, including BLC, WNT, EGFR–RAS–MAPK, PI3K (PTEN), protein kinase A, retinoblastoma (Rb), RTK/RAS/beta-catenin, and apoptosis. In the meta-analysis of nine studies by Raffone et al., PTEN loss in endometrial hyperplasia is highly related to an increased risk of EC (OR of 3.32; 95% CI 1.59–6.97, *p* = 0.001) [33]. HPV may potentially cause PTEN loss and be a part of the carcinogenesis identified in 33% of patients with HPV-positive oropharyngeal carcinoma [34].

Traditionally, the risk factors of type 1 EC are related to the “unopposed estrogen” hypothesis, which includes obesity, exogenous estrogen, and chronic hyperinsulinemia [22]. On the contrary, progesterone did not oppose the mitogenic effect of estrogen. The G-protein-coupled estrogen receptor (GPER) activated the MAPK/PI3K pathway, promoting the accelerated growth of EC cells, a mechanism similar to the role of PTEN in this process. The increased estrogen receptor alfa/beta ratio was also associated with the growth of type 1 EC [35,36]. On the other hand, studies demonstrate that HPV and estrogen play synergistic roles in the carcinogenesis of CC and oropharyngeal carcinoma [37]. ER alpha and apolipoprotein B mRNA-editing catalytic polypeptide 3 (APOBEC3, A3) increased with HPV infection, and A3 could increase the integration of HPV DNA in the host. Therefore, HPV and estrogen could be cofactors in the pathogenesis of EC, which requires further future studies. On the other hand, type 2 tumors, especially the serous type, may be more closely linked to mutations in p53, p16, and HER2 [31]. The HPV E6 protein’s inhibition of p53 is a well-established mechanism in CC and may also have relevance in the development of type 2 EC. These findings suggest a complex interplay between HPV, estrogen, and genetic mutations in different types of EC, warranting further investigation.

To the best of our knowledge, this study represents the initial endeavor to establish a connection between HPV and uterine corpus cancer, particularly endometrioid endometrial cancer, through two extensive database sets. Having employed a nationwide population-based approach coupled with data from the national cancer registry, this double-database cohort study is a robust method for scrutinizing the cancer’s etiology. Its comprehensive follow-up and avoidance of the potential biases stemming from small sample sizes and limited demographic information could bolster the reliability of the findings [38]. Population studies can also track changes over time, offering insights into the evolution of HPV prevalence and its potential impact on cancer rates. This temporal perspective is often challenging to achieve in tissue-specific studies. Our study simultaneously considered various risk factors and comorbidities that may influence HPV infection and its association with cancer. This multifactorial analysis is essential for understanding the complex interplay in real-world scenarios. This study offers an epidemiological perspective on the proposed theory, requiring further mechanistic investigation.

A limitation of our study pertains to the possibility of coding errors within the datasets. As gynecologists, it is often assumed that cervical cancer is associated with HPV infection, and, as a result, HPV may not have been consistently coded in the records. Our study’s coding for HPV infection was primarily based on observing papilloma. While the actual prevalence of HPV was possibly underestimated due to this approach, our findings still underscore the association between HPV and the occurrence of endometrial cancer. Furthermore, the administrative datasets utilized in our study lack specific information about the types of HPV involved, which could have varying roles in carcinogenesis. Important associated factors, including obesity and length of menstrual cycles, were also not considered. Additionally, the theory behind our study faces challenges due to the heterogeneity in reports of HPV DNA within endometrial cancer cells. More in-depth analysis of how comorbidities and other health determinants may influence HPV infection risk could provide meaningful insights. Similarly, visualizations of age-specific incidence patterns may further elucidate the relationship between HPV and endometrial cancer. Unfortunately, with the existing analytical outputs, we are limited in our ability to incorporate these elements.

For future research, it would be beneficial to design studies that employ techniques such as immunocytochemistry, DNA in situ hybridization, and next-generation sequencing to investigate the presence of HPV antigen in endometrial cancer cells. Moreover, exploring the presence of perinuclear HPV virus particles in cells with nuclear inclusions using electron microscopy could provide valuable insights into this study area [39]. Exploring the interplay between HPV’s impact on PTEN mutation through the PI3K pathway, the E6 gene’s effect on p53, and the E7 gene’s influence on Rb in patients with EC holds promise for future research. This investigation could shed light on the intricate mechanisms underlying EC development. Furthermore, the potential for HPV vaccination to prevent uterine EC, as proposed by this theory, warrants attention. In the future, it would be worthwhile to investigate the effectiveness of HPV vaccines in preventing EC.

## 5. Conclusions

This nationwide population-based study demonstrates a concerning pattern of increased EC incidence associated with HPV infection among Taiwanese women. The results highlight a need for more mechanistic studies while contributing complementary evidence to motivate prevention strategies targeting HPV to alleviate this cancer burden among women.

## Figures and Tables

**Figure 1 viruses-15-02314-f001:**
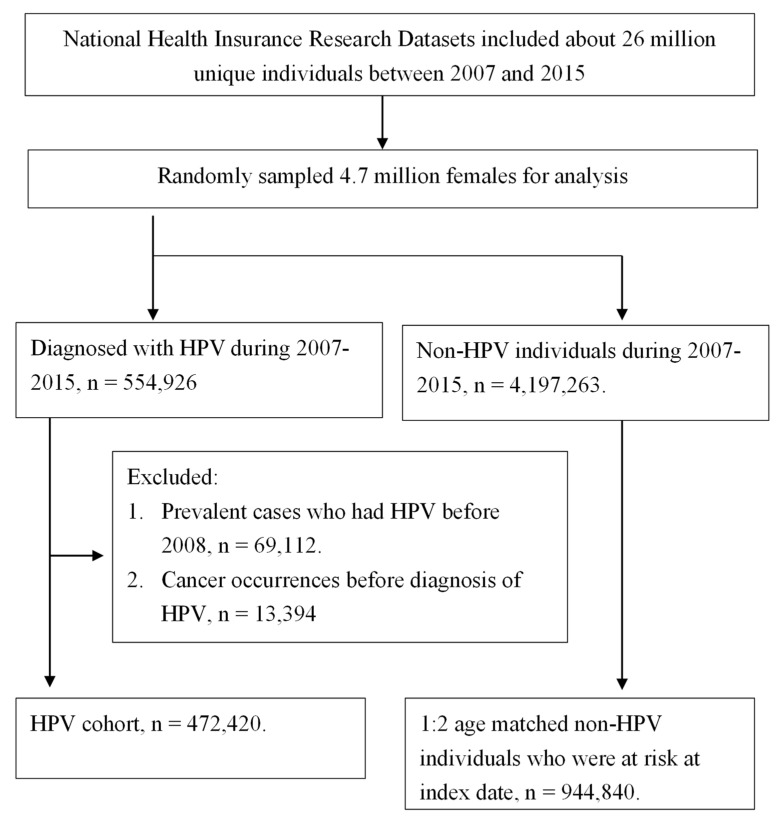
Flowchart depicting the examination of uterine body and cervical cancer incidences in correlation with HPV infection.

**Figure 2 viruses-15-02314-f002:**
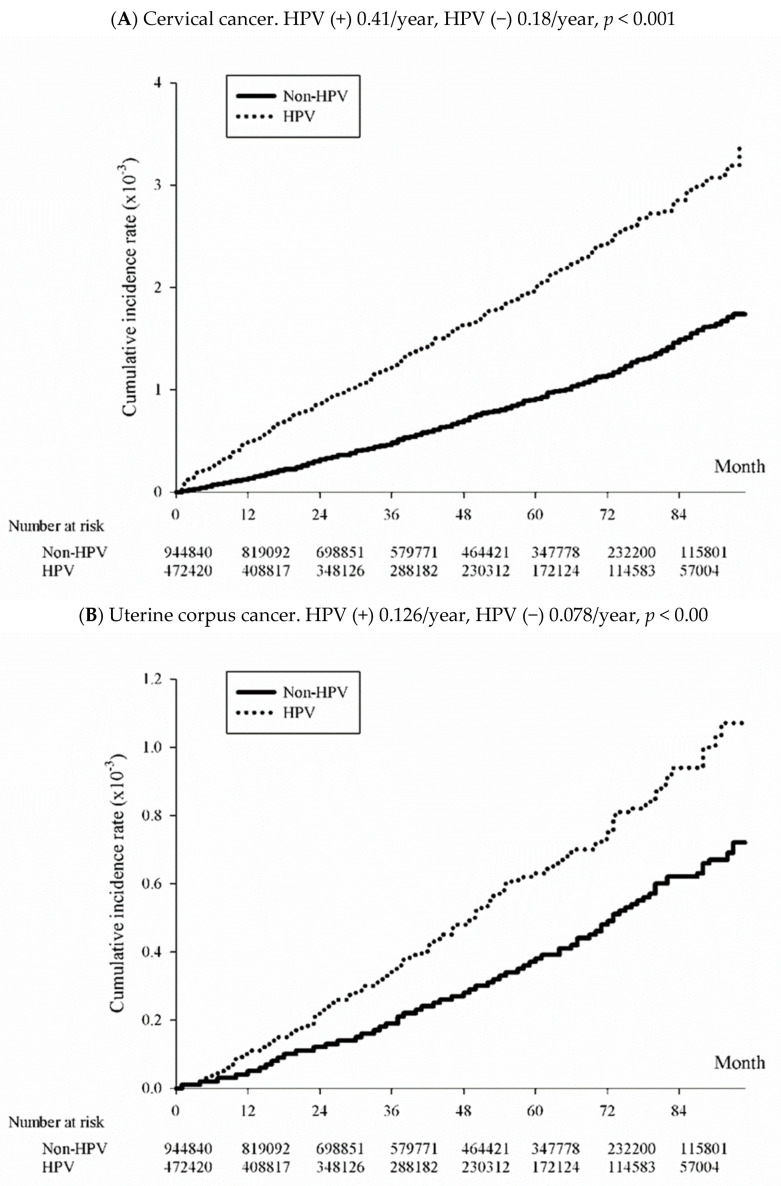
Kaplan–Meier curves of the cumulative incidence of uterine body and cervical cancers among HPV-exposed and non-HPV-exposed participants. (**A**) Cervical cancer; (**B**) all uterine corpus cancers; (**C**) type 1 endometrial cancer.

**Table 1 viruses-15-02314-t001:** Definition of coding for HPV exposure, cancer events, and comorbidity.

Variables	ICD-9 Codes Identified from Database
**Exposures**	
HPV infection	079.4, 078.1, 795.05, 795.09, 795.15, 795.19, 796.75, 796.79.
Study events	
Cancer of uterine cervix	ICD-9 codes: 180, ICD-10 code C53
Cancer of uterine corpus	ICD-9 codes: 182, ICD-10 codes C54, C55
Type 1 endometrial cancer *	814, 856, 801, 838, 848
Type 2 endometrial cancer *	857, 802, 831, 844, 898, 895, 826, 805, 845, 846
**Comorbidities**	
Ischemic heart disease	410–414
Abnormal liver function	273.4, 275.0, 275.1, 453.0, 571, 573, 576.1, 456.0, 456.1, 456.2, 789.5, 789.59, 572.2, 567, 572.4.
Renal failure	585
Chronic kidney disease	403.11, 403.91, 404.12, 404.13, 404.92, 404.93, 585, 586, 587, 274.1, 403.10, 403.90, 404.10, 404.11, 404.90, 404.91, 581, 582, 583, 590.0, 593.6, 593.9, 753.12, 753.13, 753.14
Hypertension	401–405
Diabetes mellitus	250
Lipid dysfunction	272
Stroke	430–438
COPD	490–492, 493–496
Peptic ulcer	531–533
GI bleeding	578
Gout	274

*, *p* < 0.05. ICD-O-3, International Classification of Diseases for Oncology, 3rd edition morphology code. HPV, human papillomavirus; GI, gastrointestinal; COPD, chronic obstructive pulmonary disease; ICD-9-CM, International Classification of Diseases, 9th Revision, Clinical Modification; ICD-10, International Classification of Diseases, 10th Revision.

**Table 2 viruses-15-02314-t002:** Demographic characteristics of HPV-infected and noninfected individuals.

	Non-HPV	HPV	*p*
Total case numbers	944,840	472,420	
Age			1.0000
<20	240,114 (25.41%)	120,057 (25.41%)	
20–40	343,124 (36.32%)	171,562 (36.32%)	
40–60	252,046 (26.68%)	126,023 (26.68%)	
60–80	94,532 (10.01%)	47,266 (10.01%)	
>=80	15,024 (1.59%)	7512 (1.59%)	
Residential area			<0.0001
Taipei area	344,248 (36.43%)	200,491 (42.44%)	
Northern	136,141 (14.41%)	61,490 (13.02%)	
Central	174,021 (18.42%)	90,930 (19.25%)	
Southern	130,536 (13.82%)	53,853 (11.40%)	
Kaohsiung area	138,316 (14.64%)	57,888 (12.25%)	
Eastern	21,578 (2.28%)	7768 (1.64%)	
Low-income			<0.0001
Yes	10,850 (1.15%)	4461 (0.94%)	
No	933,990 (98.85%)	467,959 (99.06%)	
Comorbidities			
Ischemic heart disease	28,420 (3.01%)	16,171 (3.42%)	<0.0001
Hypertension	97,281 (10.30%)	49,718 (10.52%)	<0.0001
Stroke	17,177 (1.82%)	8664 (1.83%)	0.5026
Diabetes mellitus	45,250 (4.79%)	20,860 (4.42%)	<0.0001
Abnormal liver function	37,355 (3.95%)	17,822 (3.77%)	<0.0001
Renal failure	6083 (0.64%)	2892 (0.61%)	0.0252
Chronic kidney diseases	10,685 (1.13%)	5531 (1.17%)	0.0352
GI bleeding	4141 (0.44%)	2195 (0.46%)	0.0266
Hyperlipidemia	65,605 (6.94%)	38,553 (8.16%)	<0.0001
COPD	11,597 (1.23%)	6454 (1.37%)	<0.0001
Peptic ulcer	63,365 (6.71%)	37,381 (7.91%)	<0.0001
Gout	9541 (1.01%)	4979 (1.05%)	0.0139

**Table 3 viruses-15-02314-t003:** Incidence of uterine and cervical cancer among HPV-exposed and unexposed individuals.

	HPV		Non-HPV			
	Event	Incidence Rate ^†^	Event	Incidence Rate ^†^	Crude HR	Adjusted HR
CC	777	41.65 (38.82–44.68)	712	18.98 (17.63–20.42)	2.195 (1.983–2.430)	2.225 (2.008–2.464)
Uterine EC	235	12.60 (11.08–14.31)	293	7.81 (6.96–8.76)	1.614 (1.360–1.917)	1.588 (1.335–1.888)
Type 1 EC	192	0.86 (0.74–0.99)	225	0.50 (0.44–0.57)	1.717 (1.417–2.082)	1.671 (1.376–2.029)
*Endometrioid adenocarcinoma*	177	0.79 (0.68–0.92)	206	0.46 (0.40–0.52)	1.729 (1.415–2.114)	1.686 (1.377–2.065)
*Adenocarcinoma, others*	14	0.06 (0.04–0.11)	18	0.04 (0.03–0.06)	1.563 (0.777–3.142)	1.499 (0.742–3.030)
Type 2 EC	18	0.08 (0.05–0.13)	26	0.06 (0.04–0.08)	1.395 (0.765–2.544)	1.450 (0.791–2.656)

^†^ Crude incidence rate, per 100,000 person-years. Adjusted HR: adjusted hazard ratio; CC, cervical cancer; EC: endometrial cancer; adenocarcinoma, others: type 1 endometrial cancer except endometrioid adenocarcinoma.

**Table 4 viruses-15-02314-t004:** Factors associated with uterine endometrial and cervical cancers.

	Uterine Endometrial Cancer			Uterine Cervical Cancer		
	aHR	95% CI	*p*	aHR	95% CI	*p*
HPV infection	1.588	1.335–1.888	<0.0001 *	2.225	2.008–2.464	<0.0001 *
no infection	Reference			Reference		
Age						
<20	0.043	0.010–0.174	<0.0001 *	0.028	0.016–0.048	<0.0001 *
20–40	Reference			Reference		
40–60	6.547	5.035–8.514	<0.0001 *	1.124	1.001–1.262	0.0481 *
60–80	4.275	3.019–6.054	<0.0001 *	1.068	0.884–1.291	0.496
≥80	1.622	0.637–4.134	0.311	0.956	0.633–1.443	0.829
Low-income	0.52	0.129–2.090	0.357	1.044	0.590–1.848	0.883
Yes						
No	1	-	-	1	-	-
Comorbidity (ref: without)						
Ischemic heart disease	0.963	0.659–1.409	0.847	0.796	0.591–1.070	0.131
Hypertension	1.622	1.275–2.063	<0.0001 *	1.125	0.934–1.355	0.214
Stroke	0.781	0.442–1.379	0.394	0.922	0.636–1.336	0.667
Diabetes mellitus	1.175	0.859–1.607	0.313	1.109	0.873–1.410	0.397
Abnormal liver function	1.303	0.942–1.802	0.110	1.334	1.067–1.669	0.011 *
Renal failure	1.107	0.351–3.494	0.863	2.725	1.235–6.009	0.013 *
Chronic kidney diseases	1.185	0.487–2.882	0.708	1.001	0.497–2.016	0.998
Hyperlipidemia	1.096	0.833–1.443	0.513	0.808	0.651–1.003	0.053
COPD	0.189	0.047–0.761	0.019 *	1.109	0.738–1.665	0.619
Peptic ulcer	0.998	0.752–1.326	0.990	1.19	0.994–1.425	0.058
Gout	1.823	1.117–2.974	0.016*	1.109	0.717–1.716	0.641

* *p* < 0.05. Adjusted HR: adjusted hazard ratio. HPV, human papillomavirus; GI, gastrointestinal; COPD, chronic obstructive pulmonary disease; ICD-9-CM, International Classification of Diseases, 9th Revision, Clinical Modification; ICD-10, International Classification of Diseases, 10th Revision; ICD-O-3, International Classification of Diseases for Oncology, 3rd Edition.

## Data Availability

The data used in this research are not publicly available due to privacy and confidentiality restrictions. These data, which include patient health records and sensitive information, were accessed and analyzed under strict ethical and legal regulations. However, qualified researchers interested in accessing the data for replication or further investigation may submit requests to the National Health Insurance Administration, Ministry of Health and Welfare, Taiwan, following the established data access procedures.

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
