# Peer review of "From Infection to Malignancy: Tracing the Impact of Human Papillomavirus on Uterine Endometrial Cancer in a Nationwide Population-Based Cohort Study"

_viruses, 2023, doi:10.3390/v15122314_

Round 1
Reviewer 1 Report
Comments and Suggestions for Authors
1. The introduction does not sufficiently justify the purpose of the paper. It is stated that the relationship between HPV and uterine cancer was investigated. The study itself describes the relationship of HPV not only with endometrial cancer but also with cervical cancer. The necessity of including СС in this study should be explained.
2. In the introduction, the statistics on the incidence of hepatitis B in hepatocellular carcinoma are spelled out. It is necessary to either delete this information and proceed directly to the consideration of HPV infection, or add more examples of which viral infections influence malignant neoplasm formation.
3. References to literature sources are relevant to the study but not up-to-date. The percentage of articles in the last 5 years leaves 46%
4. The flowchart depicting the study of uterine and cervical cancer cases in correlation with HPV infection (Figure 1.) should be inserted in the Materials and Methods section.
5. The discussion says: "HPV prevalence was higher in urban areas with relatively few people from low-income families", but in Table 2, there is no information on the number of people in the family, and furthermore, HPV prevalence was higher in high-income people than in low-income people
Reviewer 2 Report
Comments and Suggestions for Authors
Interesting study looking at the effect of HPV on endometrial cancer alongside confirmed known effect on cervical cancer. As predicted, individuals diagnosed with EC had higher HPV prevalance. Additionally, the link between HPV detection, EC, and socioeconomic factors is very interesting. As mentioned by the authors, individuals in higher socioeconomic brackets are provided with more advanced cancer screening programs and thus higher detection of EC. Many future studies can be built on these findings that were obtained from database data extraction. Overall, the authors did a good job supporting their findings in the discussion. See below comment:
1. It would be interesting to see how HPV vaccination affects HPV infection along with EC disease outcomes. Would it be possible to extract vaccination status from the database?
Author Response
Thank you for the feedback on our manuscript. You raise an excellent point about examining the impact of HPV vaccination. Unfortunately, the databases we utilized do not contain information about vaccination status. However, as you suggested, analyzing vaccination effects on HPV prevalence and endometrial cancer outcomes would be a very meaningful direction for future work. I agree this is an important factor to investigate as vaccination programs are scaled up over time. Since we are unable to address it within the constraints of this study, I can add a statement in the Discussion acknowledging this limitation and identifying vaccination impact as a worthwhile area for future research. Please let me know if you have any other suggestions for improving the manuscript. Your comments are greatly appreciated.
Reviewer 3 Report
Comments and Suggestions for Authors
It would be very interesting to correlate these findings with HPV-dependent pathology of the lower genital tract, cervix, vagina, and vulva.
Author Response
Assessing HPV-related pathology of the lower genital tract is an excellent suggestion. Due to limitations of the database variables, we were unable to examine this factor in our current study but agree it merits investigation in future work.
Reviewer 4 Report
Comments and Suggestions for Authors
The article contains an interesting multiparametric analysis of a large array of heterogeneous statistics on the characteristics of Taiwan women, covering the majority of the population. Such data sets are a valuable source of information on possible links between different parameters affecting human health.
One of the main conclusions of the article is that the analyzed data indicate a possible relationship of the HPV infection and type 1 endometrial cancer. No available literature data report any facts supporting such a causal relationship. The authors infer the possibility of this association, relying only on the fact that the incidence of endometrial cancer slightly increased in the HPV-infected cohort compared to the non-HPV cohort. However, the same study showed that the two cohorts differed significantly in a number of characteristics. It would be interesting if the authors assessed how much the risk of HPV infection, per se, depends on the presence of other factors about which information was available.
It is now generally accepted that endometrial cancer is based on its pronounced hormone dependence. Nevertheless, many factors are involved in the carcinogenesis of endometrial neoplasms, including genetic and epigenetic disorders, as well as risk factors, which include nutritional, hormonal, and hereditary causes. A number of these causes may also mediate an increased risk of persistent HPV infection, whereby the incidence of HPV infection and the risk of type 1 endometrial cancer may be parameters that demonstrate a correlation related to other factors and is not a consequence of a direct relationship due to some molecular mechanisms.
The unexpectedly low HPV incidence rate among cervical cancer patients (only 41%) raises questions about the completeness and representativeness of the data presented and the possible contribution of the HPV detection method. The authors claim that ‘while the actual prevalence of HPV was possibly underestimated due to this approach, our findings still underscore the association between HPV and the occurrence of endometrial cancer’. But the same circumstance can lead to a overestimation of the significance of this association as well.
Text accompaniment is almost exclusively tables that allow you to make only pairwise comparisons. In my opinion, it would be useful to graphically illustrate the features of the compared cohorts in the form of graphs or diagrams (for example, the distribution of the proportion of HPV-positive women according to age in women from different groups, the dependence of endometrial cancer incidence from age in HPV-positive and HPV-negative patients, etc).
Such a presentation could raise some additional issues worthy of discussion. Thus, existing studies have linked estrogenic hormonal action to the pathogenesis of endometrial cancer, with the exception of young patients in whom this hormonal action should not be present, but other factors such as HPV infection are typically ignored. In this regard, the potential contribution of HPV infection to endometrial cancer risk at earlier ages may be more pronounced.
Figure 2 mentioned in the text is absent in the manuscript, although it is discussed in detail.
My general conclusion - the article can be published only after a significant revision of the presentation and discussion of the data obtained.
Comments on the Quality of English LanguageThe text contains a number of typos and language errors, which, however, can be easily eliminated.
Round 2
Reviewer 1 Report
Comments and Suggestions for Authors
No comments
Author Response
I appreciate the reviewer taking the time to thoroughly read the manuscript. I take the reviewer's "no comments" at the second round of revision as an indication that the work was found to be scientifically sound overall.
Reviewer 4 Report
Comments and Suggestions for Authors
The article underwent minor changes, some text flaws were eliminated, the missing Figure was added.
Most of my considerations from the first report, which in my opinion could improve the presentation, were not accepted, it remains only to repeat two of them:
It would be interesting if the authors assessed how much the risk of HPV infection, per se, depends on the presence of other factors about which information was available.
It would be useful to graphically illustrate the features of the compared cohorts in the form of graphs or diagrams (for example, the distribution of the proportion of HPV-positive women according to age in women from different groups, the dependence of endometrial cancer incidence from age in HPV-positive and HPV-negative patients, etc).
I believe that the statement from the Discussion section "In our study, we found that HPV infection in women significantly elevated the risk of developing uterine cervical cancer" needs to be softened as the incidence of HPV infection and the risk of type 1 metric cancer may be parameters that demonstrate a correlation related to other factors.
As corrected, the article can be published.
Comments on the Quality of English Language
The article underwent minor changes, some text flaws were eliminated, the missing Figure was added.
Most of my considerations from the first report, which in my opinion could improve the presentation, were not accepted, it remains only to repeat two of them:
It would be interesting if the authors assessed how much the risk of HPV infection, per se, depends on the presence of other factors about which information was available.
It would be useful to graphically illustrate the features of the compared cohorts in the form of graphs or diagrams (for example, the distribution of the proportion of HPV-positive women according to age in women from different groups, the dependence of endometrial cancer incidence from age in HPV-positive and HPV-negative patients, etc).
I believe that the statement from the Discussion section "In our study, we found that HPV infection in women significantly elevated the risk of developing uterine cervical cancer" needs to be softened as the incidence of HPV infection and the risk of type 1 metric cancer may be parameters that demonstrate a correlation related to other factors.
As corrected, the article can be published.
